# Endothelial Dysfunction in Primary Aldosteronism

**DOI:** 10.3390/ijms20205214

**Published:** 2019-10-21

**Authors:** Zheng-Wei Chen, Cheng-Hsuan Tsai, Chien-Ting Pan, Chia-Hung Chou, Che-Wei Liao, Chi-Sheng Hung, Vin-Cent Wu, Yen-Hung Lin

**Affiliations:** 1Division of Cardiology, Department of Internal Medicine, National Taiwan University Hospital, National Taiwan University College of Medicine, Taipei 10002, Taiwan; librajohn7@hotmail.com (Z.-W.C.); chenghsuan.richard.tsai@gmail.com (C.-H.T.); pan.chienting.m@gmail.com (C.-T.P.); petrehcs@gmail.com (C.-S.H.); 2Cardiovascular center, National Taiwan University Hospital, Taipei 10002, Taiwan; 3Division of Cardiology, Department of Internal Medicine, National Taiwan University Hospital Yun-Lin Branch, Yun-Lin 64041, Taiwan; 4Division of Cardiology, Department of Internal Medicine, National Taiwan University Hospital Jin-Shan Branch, New Taipei City 20844, Taiwan; 5Department of Obstetrics and Gynecology, National Taiwan University Hospital, National Taiwan University College of Medicine, Taipei 10041, Taiwan; joan640124@yahoo.com.tw; 6Division of Cardiology, Department of Internal Medicine, National Taiwan University Hospital Hsin-Chu Branch, Hsin-Chu 30059, Taiwan; yo.ahliao@gmail.com; 7Division of Nephrology, Department of Internal Medicine, National Taiwan University Hospital, National Taiwan University College of Medicine, Taipei 10002, Taiwan; dr.vincentwu@gmail.com

**Keywords:** primary aldosteronism, endothelial dysfunction, vascular tone, inflammation, vascular remodeling, atherosclerosis, endothelial progenitor cell

## Abstract

Primary aldosteronism (PA) is characterized by excess production of aldosterone from the adrenal glands and is the most common and treatable cause of secondary hypertension. Aldosterone is a mineralocorticoid hormone that participates in the regulation of electrolyte balance, blood pressure, and tissue remodeling. The excess of aldosterone caused by PA results in an increase in cardiovascular and cerebrovascular complications, including coronary artery disease, myocardial infarction, stroke, transient ischemic attack, and even arrhythmia and heart failure. Endothelial dysfunction is a well-established fundamental cause of cardiovascular diseases and also a predictor of worse clinical outcomes. Accumulating evidence indicates that aldosterone plays an important role in the initiation and progression of endothelial dysfunction. Several mechanisms have been shown to contribute to aldosterone-induced endothelial dysfunction, including aldosterone-mediated vascular tone dysfunction, aldosterone- and endothelium-mediated vascular inflammation, aldosterone-related atherosclerosis, and vascular remodeling. These mechanisms are activated by aldosterone through genomic and nongenomic pathways in mineralocorticoid receptor-dependent and independent manners. In addition, other cells have also been shown to participate in these mechanisms. The complex interactions among endothelium, inflammatory cells, vascular smooth muscle cells and fibroblasts are crucial for aldosterone-mediated endothelial dysregulation. In this review, we discuss the association between aldosterone and endothelial function and the complex mechanisms from a molecular aspect. Furthermore, we also review current clinical research of endothelial dysfunction in patients with PA.

## 1. Introduction

Primary aldosteronism (PA) is now the most common and treatable cause of secondary hypertension [1], with a reported incidence ranging from 5%–15% in hypertensive patients [2]. The aldosterone excess caused by PA leads to greater increases in cardiovascular complications including coronary artery disease, myocardial infarction, stroke, transient ischemic attack, atrial fibrillation and heart failure compared to essential hypertension (EH) [3,4,5,6,7,8,9]. In addition, several cardiovascular structural and functional changes are associated with PA, including a higher percentage of left ventricular hypertrophy (LVH) [3], diastolic and subclinical systolic dysfunction [10], and increased arterial wall stiffness [11,12]. Endothelial dysfunction is a well-established fundamental cause of cardiovascular diseases and a predictor of cardiac events [13] and increasing evidence has shown that aldosterone plays an important role in the development of endothelial dysfunction. In current evidence, the effect of aldosterone is largely mediated through the mineralocorticoid receptor (MR). After binding to the MR, the aldosterone–MR complex can further translocate to the nucleus to regulate gene expression (genomic pathway) [14]. Aldosterone may also exert its rapid non-genomic effect independent of transcription or translation in a few minutes. [14]. The non-genomic effect may be mediated by the MR or other receptors (e.g., G protein-coupled estrogen receptor-1 or angiotensin receptor type 1) [14]. In addition, aldosterone can also influence vascular smooth muscle cell and endothelium function without involving MRs, either by aldosterone itself or through other receptors [15]. In this review, we discuss the association between aldosterone and endothelial dysfunction from a molecular aspect of vascular tone, inflammation response, early atherosclerosis and vascular remodeling (Figure 1). We also present an up-to-date review of clinical research on the association between PA and endothelial dysfunction.

## 2. Effect of Aldosterone on Vascular Tone

### 2.1. Vasomotor Regulation and the eNOS System

The endothelium plays an important role in the regulation of cardiovascular function, including vascular tone, vasculature and cellular activity. Nitric oxide (NO) was first reported to be a major endothelium-derived relaxing factor by Furchgott, Ignarro and Murad in the 1980s [16,17]. NO is synthesized from L-arginine in the presence of cofactors including tetrahydrobiopterin (BH4) via endothelial NO synthase (eNOS), which is an enzyme expressed in endothelial cells [18]. Released NO diffuses to vascular smooth muscle cells of the media and regulates vascular tone in three major signaling pathways. First, it activates soluble guanylate cyclase (sGC), which further leads to the formation of cyclic guanosine monophosphate (cGMP) [19]. cGMP activates protein kinase G (PKG), which hinders the calcium influx from voltage-dependent calcium channels (VDCC) and calcium release, and is mediated by inositol 1,4,5-trisphosphate (IP3) receptors [20,21]. PKG also promotes the reuptake of cytosolic calcium into the sarcoplasmic reticulum (SR) via sarco/endoplasmic reticulum calcium ATPase (SERCA). Consequently, the decrease in intracellular calcium concentration and inactivated calmodulin are unable to activate myosin light chain kinase (MLCK), which mediates vasodilation [22,23]. Second, under hypoxic status, sGC produces inosine 3′,5′-cyclic monophosphate (cIMP) in some blood vessels (especially coronary arteries) instead of cGMP [24,25], which increases vasoconstriction to hypoxia via increasing Rho-associated protein kinase (ROCK) activity. Third, NO mediates protein S-nitrosylation, including SERCA [26], G protein-coupled receptors (GPCR) [27], beta-arrestin [28] and G protein-coupled receptor kinase 2 (GRK2) [29], to regulate the vasomotor system.

The pathways involved in the activation of eNOS can be classified as being calcium-dependent or -independent. In the calcium-dependent pathway, agonists including acetylcholine [30], bradykinin [31] and histamine [32] activate receptors on the endothelial membrane to increase the intracellular calcium concentration, which then binds to calmodulin to increase eNOS activity [33]. In addition, growth factor, hormone and hemodynamic shear force [34,35] also regulate eNOS through the phosphorylation of different enzyme sites independently of the calcium concentration [36]. Both Ser635 [37] and Ser1177 (or Ser1179) [38] are sites of activation, while Thr495 [39] is inhibitory.

### 2.2. Direct Effect of Aldosterone on eNOS System Regulation and NO Production

Aldosterone influences the vasomotor system through multiple aspects. It causes the deficiency of cofactor BH4, which leads to uncoupling of eNOS and increased formation of reactive oxygen species (ROS) via a MR-dependent pathway [40]. A previous in vivo study also demonstrated that an adequate BH4 level is crucial for calcium-dependent NO formation, and that it regulates the ratio between superoxide and NO [41]. Aldosterone also dephosphorylates eNOS (Ser1177) and further decreases its activity [40]. Moreover, aldosterone itself increases the production of endothelial ROS and reduces ROS scavenging capacity, which further reduces the bioavailability of NO [42].

### 2.3. Aldosterone Induces Vasoconstrictors

Aldosterone increases the formation of vasoconstrictors. In a rat study, chronic aldosterone administration was shown to cause reduced vasodilatation to acetylcholine in the aorta through the activation of cyclooxygenase-2 (COX-2) via prostacyclin in both normotensive and hypertensive groups [43]. A similar effect of chronic aldosterone treatment has also been observed on endothelial function of mesenteric resistance arteries via increasing COX-2-derived prostacyclin and thromboxane A2 [44]. Another rat study also found that endothelin-1 played a role in blood pressure elevation and vascular hypertrophy in aldosterone- and salt-induced hypertension [45]. Endothelin-1 activates ET_A_ receptors, and this causes vasoconstriction by impairing endothelial NO synthesis and its direct vasoconstrictor effect [46].

### 2.4. Other Pathways Involving Aldosterone-Related Vasomotor Dysregulation

Aldosterone influences the vasomotor system not only via MR expressed on vascular endothelial and smooth muscle cells, but also via epidermal growth factor receptor (EGFR) and G protein-coupled estrogen receptor (GPER), with downstream mediators such as cyclooxygenase, glucose-6-phosphate dehydrogenase (G6PD) and RhoA/Rho-associated kinase [47].

#### 2.4.1. EGFR

Griol-Charhbili et al. first reported the role of the EGFR in endothelial dysfunction [48]. In their genetic mouse model (waved-2 mouse), the partial loss of EGFR tyrosine kinase activity resulted in decreased aldosterone-enhanced vasoconstriction with phenylephrine and angiotensin II, but with no influence on arterial wall remodeling [48].

#### 2.4.2. GPER

Recently, Gros et al. proposed a new receptor, which is also involved in rapid non-genomic signaling pathways, GPER (previously known as G protein-coupled receptor 30, GPR30) [49]. GPER is a 7-transmembrane-spanning estrogen receptor that was initially characterized as an estrogen-specific receptor. They also demonstrated that aldosterone causes rapid ERK phosphorylation, which mediates pro-apoptotic and anti-proliferative effects in cultured endothelial cells via GPER activation [50]. Furthermore, they demonstrated that aldosterone-mediated vasodilation was present in isolated vascular ring segments via a GPER signaling pathway [50]. These findings provide a new therapeutic strategy toward GPER pathway to modulate aldosterone-induced endothelial dysfunction.

#### 2.4.3. G6PD

Another mechanism of aldosterone-related endothelial dysfunction was proposed by Leopold et al., in which the decreased expression of G6PD regulates vascular function by limiting oxidative stress to preserve NO bioavailability [51]. They demonstrated that aldosterone infusion (10^−9^ to 10^−7^ mol/L) for 12–48 h inhibited G6PD expression in both bovine aortic endothelial cells and human coronary artery endothelial cells. Furthermore, in a mouse model, they also showed that aldosterone impaired endothelial function and vascular G6PD expression, which could then be restored by spironolactone and vascular G6PD gene transfer.

#### 2.4.4. RhoA/Rho-Associated Kinase Pathway (ROCK)

The small GTPase RhoA and its mediator Rho kinase are known to be involved in the regulation of cell motility, cell migration and smooth muscle contraction [52]. Increased Rho-associated kinase activity reduces NO bioavailability by inhibiting the expression and activation of eNOS [53], which further impairs vascular tone by unbalancing the production of vasodilators and vasoconstrictors [54]. Matsumoto et al. found lower flow-mediated dilation (FMD) and higher ROCK activity in patients with aldosterone-producing adenoma (APA) compared to those with idiopathic hyperaldosteronism (IHA) and EH [55]. In addition, ROCK activity was correlated with age, plasma aldosterone concentration and aldosterone/renin ratio. Adrenalectomy restored FMD and ROCK activity in the APA patients [55]. Kishimoto et al. also reported that 12 weeks of eplerenone treatment improved vascular function and decreased ROCK activity in IHA patients [56]. As a result, it can be inferred that the ROCK pathway may play a role in vascular function impairment in PA patients.

### 2.5. Nongenomic Effects of Aldosterone on Vascular Tones

With regards to the direct nongenomic effect of aldosterone, Romagni et al. found a rapid vasoconstrictive response to aldosterone (0.9 ng·min^−1^) in healthy volunteers who had reduced forearm blood flow after acute aldosterone exposure [57]. Another study by Gunaruwan et al. found that local intra-arterial infusion of aldosterone (10, 50, and 100 ng·min^−1^) showed no acute influence on forearm blood flow and vascular resistance in healthy volunteers [58]. However, Schmidt et al. [59] found that forearm blood flow increased under higher concentration aldosterone infusion (500 ng·min^−1^) in healthy volunteers. A series of examinations using acetylcholine (testing endothelium-dependent vasodilation), N^G^-monomethyl-l-arginine (L-NMMA, testing NO bioavailability), sodium nitroprusside (testing endothelium-independent vasodilation), and phenylephrine (which caused vasoconstriction through α1-adrenoceptors) were performed in that study. They proposed that aldosterone infusion (500 ng·min^−1^) causes rapid nongenomic effects which included increasing NO release by endothelial cells and promoting vasoconstriction by vascular smooth muscle cells [59]. The diversity of aldosterone responses may be due to its interaction with the adrenergic system [60], differences in the vascular bed and various dosages of aldosterone infusion. Schmitt et al. summarized that vasoconstriction was induced with a lower level of aldosterone, whereas vasodilatation could be found with a higher level of aldosterone [61]. However, further studies are needed to confirm the “real” action of different aldosterone levels on endothelial cells and vascular smooth muscle cells.

## 3. Effect of Aldosterone on Endothelium-Mediated Vascular Inflammation

Chronic low-grade inflammation leads to tissue destruction, proliferation, and fibrosis. Interactions between the cell surface, extracellular matrix, and proinflammatory mediators during inflammatory processes are complex [62]. The endothelium mediates the inflammatory processes through several mechanisms, and the generation of aldosterone-induced ROS plays an important role in the aldosterone mediated endothelium dysfunction [63,64].

### 3.1. ROS Systems in Vascular Inflammation

ROS, including superoxide and hydrogen peroxide, are formed by the univalent reduction of oxygen. This reaction is mediated by several enzyme systems including nicotinamide adenine dinucleotide phosphate (NADPH) oxidases and xanthine oxidase [65]. Aldosterone was shown to increase ROS production and proinflammatory transcription factors such as activator protein (AP)-1 and nuclear factor kappa B (NFκB) in a transgenic rat model [66]. The administration of antioxidant drugs such as the superoxide dismutase mimetic TEMPOL (4-hydroxy-2,2,6,6-tetramethylpiperidinyl-1-oxyl), the NADPH oxidase inhibitor apocynin, and N-acetylcysteine has been shown to decrease inflammation and injury in aldosterone-treated rodents [67,68,69].

The stimulation of ROS by aldosterone can be explained by several mechanisms. First, aldosterone and MRs play important roles in the production of ROS by NADPH oxidase [40,70]. Second, aldosterone was shown to induce oxidative stress and endothelial dysfunction by decreasing the endothelial expression of G6PD, which then reduced oxidized nicotinamide adenine dinucleotide phosphate to NADPH and thereby increased the accumulation of ROS in aortic endothelial cells in a mice model [51]. Third, the mitochondria are also involved in ROS production. Initially, ROS can open mitochondrial ATP-dependent potassium channels (mitoK_ATP_), with the subsequent enhanced production of mitochondrial ROS through the electron transport chain as an “ROS-induced ROS release mechanism” [71]. However, the current evidences of aldosterone-induced mitochondrial dysfunction in endothelium are still limited and further studies are required.

The mechanisms involved in aldosterone-induced ROS generation are thought to be via MR-independent and MR-dependent processes. Studies on fibroblasts derived from MR-deficient mice and MR-transfected human embryonic kidney (HEK) cells support the MR-independent mechanism of aldosterone-induced ROS generation via extracellular-signal-regulated kinase (ERK) 1/2, c-Jun N-terminal kinase (JNK) and GPER pathways [49,72,73].

In the NADPH oxidase-related pathway, aldosterone was shown to stimulate the expressions of NADPH oxidase 2 (NOX2/gp91phox) and p22phox through an MR-dependent mechanism and stimulate the expression of p47phox through both angiotensin type 1 (AT1)-receptor-dependent and MR-dependent mechanisms in rat aorta [74]. Leopold et al. reported that the decrease in endothelial GP6D expression caused by aldosterone could be abrogated by an MR antagonist, which also supports that it is an MR-dependent mechanism [51]. The release of ROS from the mitochondria has been speculated to be through MR-dependent and non-genomic mechanisms. Aldosterone first activates MR-dependent EGFR transactivation, and then opens mitoK_ATP_ through the phosphoinositide-3-kinase/protein kinase B (PI3K/Akt) pathway, NO synthase (NOS), and cyclic C-GMP-dependent PKG, which cause the generation of ROS from the mitochondria in the myocardium [71,75].

### 3.2. MR-Dependent and -Independent Pathways in Inflammation

Aldosterone promotes inflammation via MR-dependent and MR-independent mechanisms. Aldosterone and MR activation cause inflammation, fibrosis, and remodeling of the heart and vasculature [76,77].

#### 3.2.1. MR-Dependent Mechanism

Several rat model studies have demonstrated that MR activation causes perivascular and interstitial fibrosis in the heart, as well as aortic fibrosis and remodeling [76,77,78,79,80]. Inflammation plays a critical role in initiating fibrosis and remodeling after aldosterone infusion or MR activation. Many antioxidants have been shown to decrease inflammation and injury in rodent models [81,82,83]. Aldosterone has also been shown to increase oxidative stress in cerebral arteries, which then increases superoxide production, and the mRNA expressions of the pro-inflammatory cytokines chemokine ligand (CCL) 7, CCL8 and interleukin (IL)-1β in the brain. These effects have been shown to be abolished by spironolactone treatment and in endothelial cell MR-deficient mice [84].

High salt intake can potentiate the MR-dependent pathway. Rocha et al. demonstrated the overexpressions of intercellular adhesion molecule (ICAM), COX-2, osteopontin, and monocyte chemoattractant protein (MCP)-1 in an aldosterone-infusion and high salt intake-induced hypertension rat model, and that its effects were abolished by MR blockade [85]. A possible mechanism is that the activation of the Rho family member Rac1, a regulatory subunit of reduced NADPH oxidase, through high salt intake could activate the MR [86]. High salt intake has also been shown to enable the effects of aldosterone on oxidative stress, inflammation, and fibrosis in whole animal models through a potassium-independent mechanism [87].

#### 3.2.2. MR-Independent Mechanism

Aldosterone can also exert rapid nongenomic effects that are not blocked by MR inhibitors. GPER plays an important role in the nongenomic effect of aldosterone. Aldosterone has been shown to increase ERK phosphorylation in rat aortic vascular endothelial cells through the GPER pathway [50]. This ERK phosphorylation then increases inflammatory gene expressions and the production of pro-inflammatory cytokines [88]. In addition, ERK1/2 is activated by both aldosterone and angiotensin II. Angiotensin-receptor blockade has been shown to prevent the rapid phosphorylation of ERK1/2 in vascular smooth muscle cells, but not MR inhibitors [89,90]. The interaction between aldosterone and angiotensin II is important with regards to the rapid nongenomic effect. Lemarie et al. [91] demonstrated that the activation of NFκB requires both AT1 receptors and MRs. The activation of ERK1/2 is through AT1 receptors (MR-independent pathway), however the activation of JNK needs both AT1 and MR [91]. The activation of ERK has been shown to result in the expressions of proinflammatory transcription factors such as NFκB, adhesion factors such as ICAM-1, and chemokines such as MCP-1 [92,93].

### 3.3. Aldosterone Mediates Interactions between Endothelial and Inflammatory Cells

Endothelial cells regulate inflammatory cell infiltration and adhesion via an MR-dependent mechanism. Aldosterone increases the expressions of ICAM1, vascular cell adhesion molecule (VCAM)-1 and inflammatory markers such as COX-2 and MCP-1 in the endothelium, which induces monocytes and macrophage infiltration [85,94,95]. NFκB induces the production of adhesion molecules and chemokines [93]. MR activation increases ICAM-1 gene transcription, resulting in increased ICAM-1 surface protein and ICAM-1-dependent leukocyte adhesion to human coronary endothelial cells, which has been shown to be inhibited by spironolactone and MR knock-down with small interfering RNA [95]. An increased expression of endothelin may also contribute to oxidative stress and increased expression of adhesion molecules during mineralocorticoid excess [96,97]. Jeong et al. also demonstrated that aldosterone stimulated the release of von Willebrand factor (VWF) and IL-18 from human aortic endothelial cells [98]. In addition, aldosterone and MR activation also increase endothelial permeability. Kirsch et al. [99] reported that aldosterone caused a rearrangement of F-actin cytoskeletal fibers in human umbilical vein endothelial cells in MR- and mitogen-activated protein kinase (MAPK)/ERK-dependent mechanisms. The rearrangement of cytoskeletal fibers has been shown to increase permeability to labeled albumin through endothelial cells [99], thereby potentiating immune cell infiltration. The infiltrated monocyte-derived macrophages which are rich in NADPH oxidase then amplify the generation of ROS and worsen vascular inflammation [100]. Furthermore, Dragoni et al. reported that the engagement of ICAM-1 by inflammatory cells or antibodies also activated MAPK pathways including ERK, p38, and JNK in the endothelium, which contributed to the long-term inflammatory response [101].

### 3.4. Aldosterone Induces Systemic Inflammation Mediated by the Endothelium

As a metabolic and endocrine organ, the endothelium actively and reactively participates in hemostasis, growth factors, vascular tone and immune and inflammatory reactions [102,103]. The activation of MRs has been associated with increased systemic inflammatory markers. Our study group demonstrated that aldosterone significantly induces IL-6 protein and mRNA production in human umbilical vein endothelial cells through the MR/PI3K/Akt/NF-kB pathway [104]. Sturgis et al. also demonstrated an increase in plasma IL-6 level in a deoxycorticosterone acetate (DOCA)-salt hypertension mice model [105]. Moreover, Luther et al. reported that circulating IL-6 concentrations increased in humans after 12 h of aldosterone infusion in healthy volunteers [106]. Our study group found a significant increase in IL-6 in patients with PA compared to patients with EH [104]. In addition, Lim et al. reported that the increase in inflammatory markers related to oxidative stress in PA was associated with diastolic dysfunction [107].

MR antagonists can also help to elucidate the effect of endogenous aldosterone or MR activation on oxidative stress and inflammatory biomarkers. The expressions of systemic inflammatory markers including IL-1β, IL-6, and the NFκB subunit p105 have been observed in hypertensive rats, and the elevation of inflammatory markers has been shown to be abolished by eplerenone but not by triple antihypertensive agents [108]. These findings suggest that MR-dependent NFκB activation and its downstream actions play important roles in the expression of systemic inflammatory markers.

## 4. Effect of Aldosterone on Early Atherosclerosis

Endothelial dysfunction plays a pivotal role in the development of atherosclerosis [109]. It impairs the vasomotor system and leads to a pro-inflammatory, proliferative, and procoagulatory physiological environment, which then contributes to all stages of atherosclerosis [110]. An increasing number of clinical studies have highlighted the crucial role of aldosterone in the progression of atherosclerosis [111]. The activation of MRs in endothelial cells increases the infiltration of inflammatory cells [112], and these inflammatory cells then promote inflammation and fibrosis. The infiltrated macrophages ingest oxidized low-density lipoproteins and become foam cells, which potentiate the formation of atherosclerosis [113].

The effect of aldosterone on the formation of atherosclerotic plaque has been demonstrated in various studies. In 1995, Van Belle et al. demonstrated that aldosterone enhanced neointimal thickening after balloon denudation in a rabbit model, and that this was inhibited by spironolactone [114]. Keidar et al. demonstrated that aldosterone administration increased macrophage oxidative stress via NADPH oxidase activation and atherosclerotic lesion development in apolipoprotein-E (ApoE) knockout mice, and that the effect could be reduced by eplerenone [115]. Subsequently, Keidar et al. showed that cotreatment with an MR antagonist and tissue angiotensin-converting enzyme (ACE) and/or an angiotensin receptor-1 inhibitor resulted in better blockade of oxidative stress and further reduced the proatherogenic effect [116]. In addition, Suzuki et al. proposed that dual Renin–Angiotensin–Aldosterone system (RAAS) blockade with an MR antagonist and ACE inhibitor/angiotensin receptor blocker was more effective than single blockade [117]. Moreover, Imanishi et al. showed that a combination of eplerenone and an ACE inhibitor not only improved NO bioavailability but also decreased atherosclerotic changes in a rabbit model [118]. Taken together, these findings suggest that aldosterone-induced atherosclerosis may, in part, be independent of the angiotensin II pathway.

Marzolla et al. demonstrated the pro-atherogenic effect of aldosterone in ApoE knockout mice via an increased expression of ICAM-1 through endothelial MRs [119]. McGraw et al. also found that aldosterone increased early atherosclerosis and promoted inflammatory plaque in ApoE knockout mice via placental growth factor (PlGF) signaling [120]. PlGF binds to vascular endothelial growth factor (VEGF) type 1 receptors on endothelial and inflammatory cells, and further promotes vascular smooth cell proliferation and monocyte chemotaxis, which are fundamental processes of atherosclerosis.

## 5. Effect of Aldosterone on Vascular Remodeling

Vascular remodeling occurs when the endothelium is insulted, and this pathological response also contributes to vascular ischemic events. The endothelium can be damaged by several conditions, including smoking, diabetes, metabolic syndrome, hypertension, and other mechanical injuries such as balloon angioplasty. Aldosterone promotes vascular remodeling via MR in both vascular smooth cells and endothelial cells [121].

Many previous studies have emphasized the role of aldosterone in vascular remodeling associated with endothelial damage. Wakabayashi et al. demonstrated that eplerenone decreased collagen accumulation and fibrosis, and further inhibited neointimal hyperplasia after coronary stent deployment in a swine model [122]. Pu et al. also showed increased vascular remodeling of small arteries (increased collagen, fibronectin and ICAM-1 in the vessel wall) in aldosterone-treated rats [123]. Both spironolactone and an endothelin antagonist reversed this effect. Lacolley et al. showed increased elastin, collagen and fibronectin in artery walls, implying the structural change of large vessels in rats with hyperaldosteronism [77].

Various mediators participate in MR-related vascular remodeling, including angiotensin II, endothelin-1, EGFR and platelet-derived growth factor (PDGF). Most of these mediators activate receptors located in the walls of vascular smooth muscle cells. The role of vascular smooth muscle cell MRs in vascular remodeling has been well studied and established [121]. In this review, we focused on the functions of endothelial MRs in vascular remodeling.

Nguyen et al. demonstrated the role of endothelial MRs, in which enhanced MR activation in the endothelium increased blood pressure and vascular response to vasoconstrictors independently of MR activation in vascular smooth muscle cells [124]. However, no morphological changes were observed between endothelial cell-specific MR overexpression (ECMROE) mice and littermate control mice. In addition, Schäfer et al. demonstrated that ECMR knockout (ECMRKO) mice did not exhibit endothelial dysfunction induced by aldosterone infusion [125]. Moreover, Rickard et al. found that ECMRKO only reversed endothelial dysfunction caused by aldosterone infusion in large aorta, but not in resistance vessels such as mesenteric arteries [126], which indicated the various functions of ECMRs in different arteries. In this study, the independent role of ECMRs in the aldosterone-induced inflammatory response was observed.

## 6. Effect of Aldosterone on Endothelial Progenitor Cells

Bone marrow-derived endothelial progenitor cells (EPC) play an important role in vascular endothelium repair and serve as a biological marker for vascular function, and even as predictors of cardiovascular risk [127,128].

In 2008, Verhovez et al. demonstrated that the growth characteristics of EPC was not influenced by a high aldosterone level (both EPC from PA patients and in vitro aldosterone-treated EPC from healthy volunteers) [129]. In addition, Thum et al. demonstrated that aldosterone induces the translocation of MRs and impairs EPC function, including differentiation, migration, and proliferation [130]. In a mice study, aldosterone impaired EPC homing to vascular structures and vascularization capacity in an MR-dependent manner. In addition, reduced EPC migratory potential was demonstrated in PA patients, which could be reversed by MR blockade. In their study, there were no differences in the number of circulating EPC between a hyperaldosteronism animal model and clinical PA patients compared with control groups. Thum et al. [130] inferred that aldosterone exerts a qualitative rather than quantitative effect on circulating EPC. However, our group showed that PA patients had a lower number of circulating EPC and endothelial colony-forming units compared with EH patients [131]. We also showed that high-dose aldosterone attenuated the proliferation of circulating EPC and impaired angiogenesis. In addition, the decreased number of EPC could be reversed by adrenalectomy or spironolactone treatment. Moreover, we found that the preoperative number of EPC (log(EPC number percent) > −3.6) may be a valuable marker to predict the residual hypertension after adrenalectomy.

## 7. Effect of Aldosterone on Ion Channels in Endothelial Cells

Ion channels are found to be regulated by aldosterone/MR in endothelium and vascular smooth muscle cells [132]. We focus on the ion channels in endothelial cells and summarize their influence on vascular function.

### 7.1. Epithelial Sodium Channel

The epithelial sodium channel (ENaC) was previously known to regulate renal sodium reabsorption and blood pressure control by aldosterone in the distal nephron [133]. However, ENaC had been recently identified in vascular endothelial cells and the expression of ENaC was enhanced by aldosterone in an MR-dependent manner [134,135]. The up-regulation of ENaC will further cause stiff endothelial cell syndrome accompanied by endothelial dysfunction, which can be inhibited by aldosterone antagonist spironolactone [136]. Jia et al. [137] also showed the role of endothelial ENaC in the aldosterone-induced vascular stiffness and fibrosis in female mice and that the process could be inhibited by epithelial sodium channel antagonist (amiloride). Besides, ENaC α-subunit knockout mice also showed attenuated responses to aldosterone infusion [137].

### 7.2. Small Conductance Calcium-Activated Potassium Channels

Other endothelial ion channels regulated by aldosterone/MR are intermediate-conductance KCa3.1 and small-conductance KCa2.3 channels [138]. Small-conductance calcium-activated potassium channels (SKCa) includes three types of channels, KCa2.1, KCa2.2, and KCa2.3, expressed in different tissues [138].

KCa3.1 channels are mainly located on endothelial cell projections traversing the internal elastic lamina. It can be activated by calcium release from endoplasmic reticulum in response to stimulation of muscarinic acetylcholine receptors or other GPCRs [139]. The activation of KCa3.1 channels allows K^+^ to exit endothelial cells and further cause hyperpolarization of nearby vascular smooth muscle cells, which results in vasodilation [140]. KCa2.3 channels are presented in inter-endothelial junctions and co-localize with transient receptor potential channels in the caveolae [141]. KCa2.3 can also be activated by the increase of intracellular calcium induced by acetylcholine or mechanical shear force, and lead to endothelium-derived hyperpolarization and vasorelaxation as well [142].

Taylor et al. demonstrated that the suppression of KCa2.3 in transgenic mice caused significant blood pressure elevation and impaired vascular tone [143]. Köhler et al. also reported that genetic deficiency of KCa3.1 and KCa2.3 both impair endothelium-derived hyperpolarization-related vasodilation in murine models [144]. Also, NO-mediated vasodilation response was compromised in KCa2.3-deficient mice. Both resulted in severe dysregulation of blood pressure control. Previous studies focused on the role of MRs in KCa2.3 expressed on choroidal vasculature. MRA can increase KCa2.3 protein expression, which causes vasodilation in the eye vasculature [145]. As a result, it may be inferred that MRs regulate KCa2.3 similarly in resistance vessels, which contribute to vascular tone and blood pressure control. This emphasizes the role of vascular SKCa channel in aldosterone-induced endothelial dysfunction, which need further investigation.

## 8. Effect of Aldosterone on Extracellular Vesicles

Extracellular vesicles (EVs) are released from parent cells into extracellular environment. EVs are biovectors carrying cargo, including protein, enzymes, lipids and RNA, from parent cells [146]. They have gained more and more attention recently for their role in intercellular communication [147]. EVs can be classified as exosomes (40–100 nm), microvesicles (100–1000 nm) and apoptotic bodies (1–5 μm) according to their cellular origin, content and particle sizes [148]. EVs can even be biomarkers of several diseases, including, cancer, metabolic disorders, and cardiovascular diseases [149].

In 2012, Lopez et al. found that an increase in circulating EVs is associated with vascular remodeling and endothelial dysfunction in aldosterone-salt-treated rats [150]. Neves et al. also identified endothelial cell-derived microparticles as biomarkers of endothelial injury associated with vascular endothelial growth factor pathway inhibitors (VEGFi) anti-cancer treatment and mediators of endothelin-1 induced pro-inflammatory signaling in endothelial cells, which may contribute to VEGFi-related cardiotoxicity [149]. Besides, EVs directly promote endothelial cell senescence through ROS, causing vascular dysfunction when aging [151].

Recently, Burrello et al. found that PA patients had higher number of endothelium-derived EVs compared with EH patients and normotensive controls [152]. In addition, the number of circulating EVs correlated well with serum aldosterone level. In PA patients receiving unilateral adrenalectomy, the number of EVs decreased significantly [152]. Besides, PA-derived EVs promoted apoptosis and inhibited angiogenesis in vitro. Therefore, the level of circulating EVs may be an indicator of endothelial dysfunction and vascular injury in PA patients. However, whether the increase in EVs was due to the primary effect of aldosterone or aldosterone-induced endothelial dysfunction was not clear [153]. The study by Burrello et al. also showed that the effects of EVs on angiogenesis and apoptosis were likely to result from multiple signaling processes rather than from a single mediator [152].

## 9. Clinical Data and Treatment among PA and Endothelial Dysfunction

In 1993, Taddei et al. demonstrated a reduced vasodilatation effect of acetylcholine in patients with PA and EH, but a similar finding was not observed with the use of nitroprusside [154]. Farquharson et al. showed that short-term aldosterone administration reduced acetylcholine-induced vasodilation measured by forearm venous occlusion plethysmography in healthy young men, which indicates evidence of aldosterone-induced endothelial dysfunction [155]. Farquharson et al. also found that spironolactone improved forearm blood flow response to acetylcholine in chronic heart failure patients [156]. Macdonald et al. showed that endothelial function (acetylcholine-induced vasodilatation), accompanied with brain natriuretic peptide, collagen markers, and QT interval length, improved after spironolactone treatment in patients with asymptomatic or mild heart failure [157].

Previous and recent metanalyses have reported an increased risk of several cardiovascular diseases in PA patients compared to EH controls [3,8]. Endothelial dysfunction has been significantly correlated to several cardiovascular events [13]. Various evaluation tools are used to assess endothelial function, including circulating biomarkers, non-invasive physiological assessment, and invasive physiological assessments in PA patients, with non-invasive physiological tools used most and invasive assessment tools such as forearm venous occlusion plethysmography and cardiac catheterization used less [158,159,160]. However, no single method currently fully represents the various aspects of vascular endothelial biology, and a combination of different evaluation tools can help clinical researchers to measure endothelial function. Although some of these tests have been commercialized, they are currently reserved for research and to aid in clinical judgment but not for disease diagnosis. Accumulating data from several tests have been reported to be well correlated to further clinical cardiovascular disease outcomes and adverse events [160]. Studies on treatments for PA have also provided data on outcome analysis and the pathogenesis of aldosterone on blood vessels and related treatment responses, which are summarized in Table 1.

### 9.1. Circulating Biomarkers Associated with Endothelial Dysfunction

Several circulating biomarkers have been studied for endothelial dysfunction, and some have been investigated in PA patients. Asymmetric dimethylarginine (ADMA) has been reported to be a marker of endothelial dysfunction, but it has not been shown to be able to differentiate between PA patients and EH controls [161]. Inflammatory-related adhesion molecules vWF, ICAM-1, and oxidized low-density lipoprotein (ox-LDL) have been found to be significantly higher in PA patients than in EH controls [162]. Our previous study also reported an elevated serum level of the proinflammatory cytokine IL-6 and increased myocardial fibrosis among PA patients and cell studies, and the possible mechanism involved MR/PI3K/Akt/NF-kB pathways [104].

### 9.2. Flow-Mediated Dilation (FMD)

Non-invasive physiological assessments have been frequently used in studies of aldosterone-related endothelial dysfunction in PA patients. FMD currently remains the technique of choice, and it has become widely used in clinical studies with protocolized guidelines for standardized procedures with good correlations with clinical outcomes with regards to long-term mortality and cardiovascular disease [158,159,160]. Impaired endothelial dysfunction with decreased FMD has been reported in several clinical studies. Nishizaka et al. demonstrated a strong negative association between an excess of aldosterone and low FMD performance, indicating impaired endothelial function among patients with resistant hypertension later diagnosed with hyperaldosteronism [163]. Autosomal dominant polycystic kidney disease (ADPKD) patients with a higher aldosterone level diagnosed with PA have also been reported to have lower FMD performance [164]. Our previous study demonstrated impaired endothelial function with significantly lower FMD and nitrate-mediated dilation (NMD) in patients with PA than in patients with EH, with impaired vascular smooth muscle cell function and suppressed SERCA 2a expression; the impairment was restored after adrenalectomy [165]. Matsumoto et al. [55] demonstrated a significantly decreased FMD performance among patients with APA, a subtype of PA, compared to patients with idiopathic adrenal hyperplasia (IAH), another subtype of PA, and patients with EH. The decrease in FMD was significantly correlated with increased plasma aldosterone level and ROCK activity, and these parameters were all restored after adrenalectomy [55].

### 9.3. Peripheral Arterial Tonometry (PAT) to Evaluate Endothelial Dysfunction

In addition to FMD reflecting macrovascular endothelial function, reactive hyperemic index (RHI) and augmentation index (AI) performed under peripheral arterial tonometry (PAT) have been shown to indicate microvascular function. RHI measures the change of pulse arterial volume before and after upper arm cuff occlusion, which represents endothelial function; whereas AI measures the relative contribution of augmented pressure due to reflected wave, which indicates arterial stiffness [173]. In conjunction with FMD, differences between patients with different subtypes of PA have provided further clues to the pathogenesis of disease. We previously reported that PA patients had higher arterial stiffness but comparable microvascular endothelial function to EH patients with an increased AI but similar RHI [167]. Kishimoto et al. reported impaired microvascular endothelial function compared to EH patients among PA patients with IAH presenting with a decreased RHI, while PA patients with APA showed decreased RHI and FMD [166]. Matsumoto et al. also reported restored microvascular dysfunction with improved RHI and ROCK activity, and decreased aldosterone–renin ratio after eplerenone treatment in patients with IAH [56]. Different types of endothelial dysfunction are found among PA subtypes, and APA has been attributed to micro- and macrovascular endothelial dysfunction while IAH has been shown to cause microvascular endothelial dysfunction [174].

### 9.4. Pulse Wave Velocity (PWV)

PWV analysis is widely used to evaluate endothelial function based on accumulating data suggesting that endothelial dysfunction is an antecedent of artery stiffening by promoting atherosclerosis or vascular muscle contraction [175]. McEniery et al. demonstrated a significant inverse association between PWV and endothelial function assessed by FMD using a large cohort of healthy volunteers who were free of cardiovascular disease, risk factors, and medication [176], Naka et al. also reported the impaired flow-mediated reduction of PWV after endothelium-dependent pharmacological stimuli among heart failure patients compared to healthy controls [177].

Bernini et al. reported both significantly elevated femoral and radial PWV and aortic AI in PA patients compared with EH patients or normotensive controls [168]. Štrauch et al. also reported significantly higher PWV in PA patients compared to EH or normotensive individuals [169], and further reported that decreased PWV and AI were observed among PA patients after adrenalectomy, while no changes were found in those treated with spironolactone [170]. Rosa et al. also demonstrated both elevated peripheral and central PWV among PA patients compared to EH controls, and that this was correlated with plasma aldosterone level [171]. We previously reported increased vascular stiffness and early atherosclerosis presenting as increased PWV and carotid artery thickening in APA patients, and that the change could be reversed by adrenalectomy [11] within 6 months; this was correlated with baseline vascular condition, blood pressure, and hormonal changes [12]. Otherwise, as we know, the potassium voltage-gated channel subfamily J member 5 (KCNJ5) gene is the most common somatic mutation in APA patients, and the KCNJ5 mutation results in loss of potassium and entry of sodium [178]. APA patients with KCNJ5 mutation are younger and have higher aldosterone levels and more severe hypokalemia compared with non-mutant carriers [179]. Although APA patients with KCNJ5 mutation had better hypertension recovery after adrenalectomy, we observed no difference in PWV improvement after adrenalectomy regardless of whether or not the patient had the KCNJ5 mutation [172].

## 10. Conclusions

Endothelial function is crucial to maintain vascular homeostasis, including vascular tone, cellular adhesion, thromboresistance, inflammation and even immunity. The aldosterone excess in PA patients exerts detrimental effects on the endothelium, causing impairment of vascular relaxation, increased oxidative stress, vessel inflammation, vascular remodeling and early atherosclerosis. An in-depth understanding of the pathophysiology of aldosterone-induced endothelial dysfunction may inspire more therapeutic strategies aimed at improving endothelium function in a wide variety of cardiovascular diseases.

## Figures and Tables

**Figure 1 ijms-20-05214-f001:**
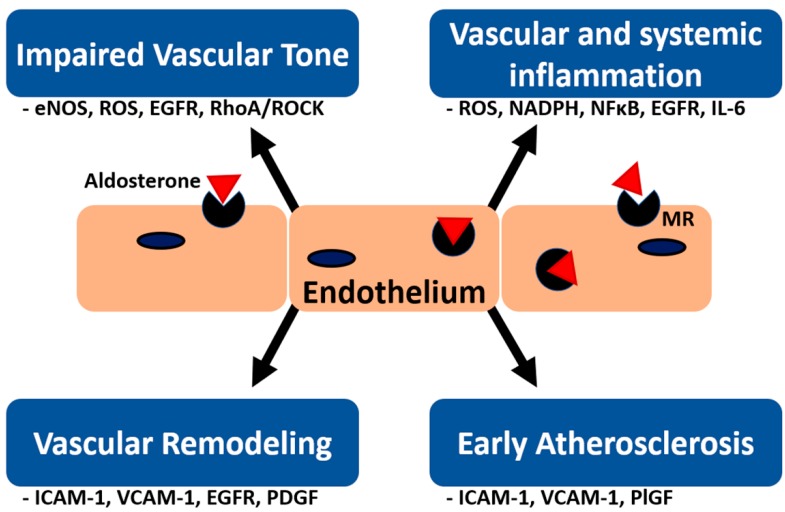
Pathophysiology of aldosterone-induced endothelial dysfunction. There are four major causes of aldosterone-induced endothelial dysfunction including impaired vascular tone, vascular and systemic inflammation, vascular remodeling and early atherosclerosis. MR = mineralocorticoid receptor; eNOS = endothelial NO synthase; ROS = reactive oxygen species; EGFR = epidermal growth factor receptor; IL = interleukin, NADPH = nicotinamide adenine dinucleotide phosphate; ICAM = intercellular adhesion molecule; VCAM = vascular cell adhesion molecule; PDGF = platelet-derived growth factor; and PIGF = placental growth factor.

**Table 1 ijms-20-05214-t001:** Summary of clinical studies on primary aldosteronism and endothelial dysfunction.

Author and Year	Assessment Method	Study Group	Control Group	Treatment	Change and Effect on Endothelial Function
Biomarker					
Verhovez et al. (2008) [129]	Biomarkers: EPC	PA	Healthy control	-	No difference between high aldosterone treated EPCs with healthy EPCs
Thum et al. (2011) [130]	Biomarkers: EPC PAT	PA	Healthy control Mouse EPCs	Spironolactone	EPCs from PA showed reduced migratory potential and reduced RHI. EPCs treated with aldosterone in vitro showed impaired multiple cellular functions through MR-dependent pathway.
Wu et al. (2011) [131]	Biomarkers: EPC PWV	PA	EH	Adrenalectomy or spironolactone	Decreased circulating EPCs and endothelial CFUs, improved after treatment
Matrozova et al. (2016) [161]	Biomarkers: ADMA	PA	EH and healthy control	-	No difference between PA and EH
Liu et al. (2014) [162]	Biomarkers: vWF	PA	EH	-	Increased
	Biomarkers: ICAM-1	PA	EH	-	Increased
	Biomarkers: ox-LDL	PA	EH	-	Increased
Chou et al. (2018) [104]	Biomarkers: IL-6	PA	EH	Adrenalectomy	Elevated IL-6 among PA using mineralocorticoid receptor/PI3K/Akt/NF-kB pathway
FMD/PAT					
Nishizaka et al. (2004) [163]	FMD + NMD	Resistant hypertension with hyperaldosteronism	Resistant hypertension without hyperaldosteronism	Spironolactone	Resistant hypertension with hyperaldosteronism showed lower FMD and improved after spironolactone.
Lai et al. (2016) [164]	FMD	ADPKD with PA	ADPKD without PA	-	ADPKD with PA shows lower FMD.
Chou et al. (2015) [165]	FMD + NMD	PA	EH	-	FMD and NMD are both decreased in PA SERCA2a suppression is observed in vascular smooth muscle.
Matsumoto et al. (2015) [55]	FMD + NMD	APA IHA	EH	Adrenalectomy on APA	FMD lower in APA than IHA and EH NMD no significant difference
Kishimoto et al. (2018) [166]	FMD + NMD PAT	APA IHA	EH	-	FMD lower in APA NMD lower in APA RHI lower in APA and IHA than EH.
Kishimoto et al. (2019) [56]	FMD + NMD PAT PWV	IHA	-	Eplerenone	RHI, NMD improved. PWV, ROCK activity decreased. No change in FMD and IMT.
Chang et al. (2015) [167]	PAT	PA	EH	-	PA had significantly higher AI but not RHI than EH
PWV					
Bernini et al. (2008) [168]	PWV	PA	EH and normotensives	-	PA showed more dysfunction and thicker IMT than EH and normotensives.
Strauch et al. (2006) [169]	PWV	PA	EH and normotensives	-	PA showed more dysfunction than EH and normotensives.
Strauch et al. (2008) [170]	PWV	PA receiving adrenalectomy	PA receiving spironolactone	Adrenalectomy or spironolactone	Endothelial dysfunction improved after adrenalectomy; not seen in spironolactone.
Rosa et al. (2012) [171]	PWV	PA	EH	-	PA showed more dysfunction
Wu et al. (2011) [131]	PWV	PA	EH	Adrenalectomy or spironolactone	Increased PWV in PA
Lin et al. (2012) [11]	PWV	APA	EH	Adrenalectomy	IMT and dysfunction improved after operation
Liao et al. (2016) [12]	PWV	APA	-	Adrenalectomy	Dysfunction improved after operation
Chang et al. (2017) [172]	PWV	APA with KCNJ5 (+)	APA with KCNJ5 (−)	Adrenalectomy	-

PA = primary aldosteronism; EH = essential hypertension; APA = aldosterone-producing adenoma; IHA = idiopathic hyperaldosteronism; EPC = endothelial progenitor cells; PAT = peripheral arterial tonometry; PWV = pulse wave velocity; CFUs = colony forming units; ADMA = asymmetric dimethylarginine; vWF = von Willebrand factor; ICAM = intercellular adhesion molecule; ox-LDL = oxidized low-density lipoprotein; IL = interleukin; FMD = flow-mediated vasodilation; NMD = nitrate-mediated dilation; ADPKD = autosomal dominant polycystic kidney disease; SERCA = sarco/endoplasmic reticulum calcium ATPase; IMT = carotid intima-media thickness; AI = augmentation index; RHI = reactive hyperemic index and KCNJ5 = potassium voltage-gated channel subfamily J member 5.

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
