# Peer review of "Endothelial Dysfunction in Primary Aldosteronism"

_ijms, 2019, doi:10.3390/ijms20205214_

Round 1

Reviewer 1 Report

In this clever review Chen et al. described the complex interactions among endothelium, inflammatory cells, and other players determining endothelial dysfunction in patients with primary aldosteronism. Reporting evidence about aldosterone effect in the development of endothelial damage, the authors further elucidate the association between endothelial dysfunction and the increased cardiovascular risk of patients with PA compared to essential hypertensives. The manuscript is well-organized and well-written, each paragraph is clear and accurate.

I have only minor comment. Extracellular vesicles (EVs) have been described as novel players in intercellular communication and potential biomarkers for cardiovascular disease. Reflecting endothelial cells functional state, EVs have been recently demonstrated to be a marker of aldosterone-associated target organ damage and endothelial dysfunction (see Lopez A et al. PLoS One 2012, Burrello J et al. Hypertension 2019, Neves et al. Hypertension 2019). EVs could be considered biomarkers, but maybe also new players, of the endothelial dysfunction in PA patients. The authors should consider adding a paragraph on this intriguing topic and developing field of the research on hyperaldosteronism.

Author Response

This point is well taken. We had added a paragraph titled “Effect of aldosterone on extracellular vesicles”. (Line 410-434)

Reviewer 2 Report

  1. This review focuses on endothelial dysfunction in primary aldosteronism (PA). Emphasis is placed on vascular aldosterone actions, particularly those mediated via the endothelial cell mineralocorticoid receptor (ECMR). These are topical areas of research and translationally relevant as hyperaldosteronism is recognized to be a major cause of secondary and resistant hypertension. While the review is potentially interesting and valuable it currently suffers in its organization, particularly if it is to be directed towards a broad audience. Specifically, the paper would be improved by an earlier discussion of the relatively recent findings of aldosterone/MR activation in vascular cells and more clearly presenting the possible contributions of genomic and non-genomic effects (including the concept of rapid actions). Then lead in to their roles in various aspects of vascular dysfunction.
  2. Further, Section 7 (clinical aspects of PA) does not directly parallel the previous sections on vascular dysfunction (for example the section on early atherosclerosis does not specifically refer to PA and Section 7 does not refer to atherosclerosis) despite the Conclusion (Section 8) implying that this has been discussed. Similarly, changes in vascular PWV and stiffness in PA are discussed in Section 7 but this is not extensively covered in early sections despite a growing literature on the role of aldosterone/ECMR in vascular stiffness (see studies from Sowers, J.; Bender, S.; Jaffe, I.Z.etc).
  3. It should be stated that while non-renal actions are being emphasized some of the studies quoted (using inhibitors) will be influenced by effects on multiple tissues as well as through changes in hemodynamics.
  4. Minor concerns include a number of instances of awkward sentence structure and grammatical errors. There are a number of repetitive sections, for example several sections begin by restating information given in previous paragraphs.  As a review it may also be enhanced by the inclusion of additional conceptual figures – the current single figure is very general.

Minor Comments/Suggestions:

5. Perhaps discussion of aldosterone effects on ion channels could be included?

6. In several places there are single sentence paragraphs. These tend to simply state the results of a study without providing context or a statement of the significance of the findings. There are a number of cases where conflicting results from the literature are mentioned (for example, aldosterone-mediated dilation and constriction). It would be of value to explain in greater depth the possible reasons and significance (granted dose-response relationships and tissue heterogeneity are briefly mentioned).

7. Line 118: GPER Line 126: It is unclear what ‘another’ relates to?

8. Lines 140 – 142: the relevance of the leukocyte studies in the context of the endothelium could be expanded? Lines 136 – 147: Section reads like a list of studies without drawing the information together.

9. Lines 156 – 158: is this correct?

10. Line 174: change to ‘rat’.

11. Line 184 – 186: are there other mitochondrial mechanisms?

12. Line 287 – 301: not that convincing as ACE inhibition gives additive effects. Are all trials consistent with this?

13. Section 5 makes little reference to PA.

14. Section 7.2: may be worth explaining some of the abbreviations (and their significance) further?

15. Line 452 – 453: significance of this particular channel (mutation) could be provided?

16. Line 457: ? homeostasis.

Author Response

    1. This point is well taken. We had added a paragraph in the “Introduction” (Line 46-53). We understand both genomic and non-genomic pathway play important role on aldosterone-induced endothelium dysfunction. However, many of them (especially non-genomic effects) are not known yet. In addition, non-genomic effects are largely studied in renal endothelium and cardiomyocyte.  The data involving vascular endothelium is very limited. Therefore, it is difficult to illustrate a full figure of detail genomic/non-genomic pathway and their downstream in vascular endothelium. Therefore, we briefly described the conception of genomic / non-genomic pathway in this article.
    2. This point is well taken. The current review “Endothelial dysfunction in Primary aldosteronism” focus on the possible “endothelial dysfunction” happening in patients with “Primary aldosteronism”.

      The excess of aldosterone in PA patients damages vascular systems in many aspects including endothelium, vascular media and adventitia.

      In cell studies, we can clearly focus on the effect on endothelium. However, it sometimes difficult to dissect which cell response the effect in animal studies. For example, aldosterone infusion causes both impaired relaxation and increased artery stiffness. We do not know exactly how much percentage through endothelial cell or smooth muscle cell, even fibroblast. The situation became worst in clinical studies. In clinical studies, we only know PA patients had worse FMD or PWV data. However, we do not know which role endothelium play in.  In addition, the clinical data is very limited comparing to bench studies. Therefore, in section 7, we try to put all studies which may associated with endothelial dysfunction in this review. Of course, the clinical studies are not parallel to bench studies.

      We noticed clinical studies in “early atherosclerosis” part is not from PA patients. We decided to delete them.

    3. This point is well taken. We understand that aldosterone had influence on multiple tissue and organ, and as well as inhibitors. The situation is inevitable in almost “all” in-vivo studies, such as animal or clinical studies. Of course, the damage by aldosterone or rescue by inhibitor may though hemodynamic effect (rise or improve of blood pressure). It is difficult to address this point in every in-vivo studies. In addition, I believed readers in this journal will understand this limitation. 

    4. Thanks for the comments. Before submission, this paper received extensively English editing by a professional English-editing company (certification). 

       This time, we had carefully reviewed and corrected the sentence structure and grammatical errors. We also found the repetitive concept in the last paragraph of section 5, and the paragraph was deleted.

      The genomic effect of aldosterone on endothelial cells increases the expressions of ICAM-1 and VCAM-1. ICAM-I[95] and VCAM-I[128] recruit leukocytes, and this further promotes an inflammatory response. In addition, aldosterone also suppresses G6PD activity, which further increases ROS generation[50]. All of these inflammatory responses contribute to subsequent vascular remodeling[121].

             It is a good idea to have more figures, and we are trying to sketch a figure to describe detail signal pathway of endothelium. However, we face a tough situation that the data is very limited in endothelium.

             For example, in early atherosclerosis and inflammation, we draw a figure (below).

          However, many down-stream pathway is un-known, and some pathways are found in other cells (such as smooth muscle cell). Although I personally believe there should be similar in endothelium, but we cannot nail it down without definite data. 

          I think it is too early to sketch the detail pathway in endothelium with current evidence. Therefore, we think it is more appropriate to choose a figure which gives general ideas.

    5. Thanks for your suggestion! We had added a paragraph titled “Effect of aldosterone on ion channels in endothelial cells”. (Line 373-409)

    6. Thanks for your comments. We had checked and revised the single sentence paragraphs. We also rewrote the paragraph about the nongenomic effects of aldosterone on vascular tones. (Line 161-168) “However, ……muscle cells.”

    7. We rewrote the sentence. (Line 133) “These findings provide a new therapeutic strategy toward GPER pathway to modulate aldosterone-induced endothelial dysfunction.”

    8. Thanks for the comments. As we mentioned previously, it is difficult to attribute the effect of leukocyte Rho-associated kinase activity to endothelium. We decided to delete this paragraph and add a summarized sentence to clarify the role of ROCK pathway.

      Leukocyte Rho-associated kinase activity has been reported to be a new biomarker of cardiovascular events[54] and to be significantly correlated with flow-mediated vasodilation (FMD)[55].

(Line 154-155)

……As a result, it can be inferred that the ROCK pathway may play a role in vascular function impairment in PA patients.

9. We rewrote this result to be more clear. (Line 161-174)

10. Sorry for the typo. We had corrected it.…..” in a transgenic rat model[66]”.(Line 186)

11. This point is well taken. There are some other mechanisms related to aldosterone induced mitochondria dysfunction and ROS production such as PPAR-γ in podocytes[1], MR-mediated, mitochondrial-originated, ROS-dependent ERK1/2 activation in renal tubular epithelial cells[2] and A-kinase anchor protein (AKAP)-12[3] in cardiac fibroblasts. We didn’t include these pathways in this article due to that the correlations between these pathways and the aldosterone-induced endothelium dysfunction are not very consolidated. In fact, the studies of aldosterone induced mitochondrial dysfunction in endothelium are extremely limited. More studies were required to confirm the results. We added this description in this paragraph.

(Line 198-200)

However, the current evidences of aldosterone induced mitochondrial dysfunction in endothelium are still limited and further studies are required.

12. This point is well taken.  Only few studies investigating the combination of MR antagonist and ACEi/ARB toward the reduction of atherosclerosis, and most of them tends to have positive results.

   However, as to the role of ACEi/ARB in lipid metabolism and atherosclerosis, some results are neutral and some showed improvement of lipid profile and atherosclerosis process[4, 5]. Also, apart from the effect of aldosterone, Weiss et al. showed the role of angiotension II in the development of atherosclerosis[6]. Recently, Lee et al. also demonstrate the combination of ARB and statin exerts synergistic anti-atherosclerotic effects[7]. All these studies may partially explain additive effect of ACEi and mineralocorticoid receptor antagonist to decrease atherosclerosis.

   Of course, we have to consider about some bias (ex. selection publication bias). However, in current evidence, the majority of studies investigating the combination of MR antagonist and ACEi/ARB on atherosclerosis tends to have positive results.

13. Thanks for your comment! In section 5, we focused on the effect of aldosterone on vascular remodeling via “endothelial cell”. As we mentioned previously, it is difficult to tell the vascular change in PA patients was mediated through either endothelium and/or vascular smooth muscle cell. We decided to put these clinical trials in Section 7 (now Section 9).

   In addition, we put the ECMR KO studies by Schafer et al. and Nguyen et al. because the studies using tissue-specific KO mouse model confirmed these effects through endothelium. They are very important studies in this review.

14. Thank you! We had added a paragraph to explain RHI and AI!

(Line 491-494)

RHI measure the changes ……which indicate arterial stiffness[168].

15. Thanks for the comments.  We rewrote this paragraph and shortly describe the significance of KCNJ5. (Line 523-527)   “Otherwise, ….”

16. Thanks for the comments. We had changed it. (Line 531)

Reference:

  1. Zhu, C.; Huang, S.; Yuan, Y.; Ding, G.; Chen, R.; Liu, B.; Yang, T.; Zhang, A., Mitochondrial dysfunction mediates aldosterone-induced podocyte damage: a therapeutic target of PPARgamma. Am J Pathol 2011, 178, (5), 2020-31.
  2. Zhang, A.; Jia, Z.; Guo, X.; Yang, T., Aldosterone induces epithelial-mesenchymal transition via ROS of mitochondrial origin. Am J Physiol Renal Physiol 2007, 293, (3), F723-31.
  3. Ibarrola, J.; Sadaba, R.; Martinez-Martinez, E.; Garcia-Pena, A.; Arrieta, V.; Alvarez, V.; Fernandez-Celis, A.; Gainza, A.; Cachofeiro, V.; Santamaria, E.; Fernandez-Irigoyen, J.; Jaisser, F.; Lopez-Andres, N., Aldosterone Impairs Mitochondrial Function in Human Cardiac Fibroblasts via A-Kinase Anchor Protein 12. Sci Rep 2018, 8, (1), 6801.
  4. Gude, D., Angiotensin-converting enzyme inhibitors in lipid metabolism and atherosclerosis: An ace up the sleeve? Journal of the Scientific Society 2014, 41, (1), 59-60.
  5. Aoyama, T.; Minatoguchi, S., [The effect of ARB on prevention of atherosclerosis]. Nihon rinsho. Japanese journal of clinical medicine 2011, 69, (1), 92-9.
  6. Weiss, D.; Taylor, W. R., Deoxycorticosterone acetate salt hypertension in apolipoprotein E-/- mice results in accelerated atherosclerosis: the role of angiotensin II. Hypertension (Dallas, Tex. : 1979) 2008, 51, (2), 218-24.
  7. Lee, S.-G.; Lee, S.-J.; Thuy, N. V. P.; Kim, J.-S.; Lee, J.-J.; Lee, O.-H.; Kim, C.-K.; Oh, J.; Park, S.; Lee, O.-H.; Kim, S. H.; Park, S.; Lee, S.-H.; Hong, S.-J.; Ahn, C.-M.; Kim, B.-K.; Ko, Y.-G.; Choi, D.; Hong, M.-K.; Jang, Y., Synergistic protective effects of a statin and an angiotensin receptor blocker for initiation and progression of atherosclerosis. PloS one 2019, 14, (5), e0215604.

Round 2

Reviewer 1 Report

Thank you to the authors for their response. All points have been answered to my satisfaction, and I recommend the manuscript for publication.

Author Response

We would like to express our thanks to all of the reviewers for the helpful questions and comments. Your excellent review and wonderful suggestion really help a lot for the improvement of our review article!

Sincerely Yours,

Yen-Hung Lin, MD, PhD, FESC

Division of Cardiology, Department of Internal Medicine

National Taiwan University Hospital

President of Taiwan Society of Aldosteronism

Head, Cardiovascular research team, TAIPAI study group

Reviewer 3 Report

The review presented by Chen et. al introduced recent studies on ECs in primary aldosteronism from physiology to pathology, which could be interest to the related researchers. The author concluded the effect of aldosterone in cardiovascular diseases including vascular inflammation, atherosclerosis and vascular remodeling. Moreover, its effects on  EC progenitors, ion channels and EC-EVs were introduced in detail. The review is well-structured and carefully organized. I do not have concerns toward it.  

Author Response

(The authors gave the same response as above.)

Reviewer 4 Report

This review covered key aspects of endothermic dysfunction in PA and has been improved after satisfactorily addressing most of the reviewers' concernssatis. I suggest including the Alsosterone signaling Figure shown in the response to comments with a note indicating that some of the downstream signaling pathways were discovered in non-ECs and further studies are needed to confirm them in ECs.

Comments:

1. Angiotensin II and AT1R should be added o Fig.1

2. Authors entered wrong line numbers in the response to comments making it difficult to locate changes.   
